# The Influence of Hop Phenolic Compounds on Dry Hopping Beer Quality

**DOI:** 10.3390/molecules27030740

**Published:** 2022-01-24

**Authors:** Irina N. Gribkova, Larisa N. Kharlamova, Irina V. Lazareva, Maxim A. Zakharov, Varvara A. Zakharova, Valery I. Kozlov

**Affiliations:** All-Russian Scientific Research Institute of Brewing, Beverage and Wine Industry—Branch of V.M. Gorbatov Federal Research Center for Food Systems, Brewing Technology Department, 119021 Moscow, Russia; harlara@yandex.ru (L.N.K.); lazirka@gmail.com (I.V.L.); mazakharoff@mail.ru (M.A.Z.); AVA-83@yandex.ru (V.A.Z.); izhlineyo@yandex.ru (V.I.K.)

**Keywords:** phenolic compounds, isoxanthogumol, hop resins, phenolic compounds, β-glucan, soluble nitrogen content, taste beer descriptors, dry hopping

## Abstract

Background: The article considers the phenolic hop compounds’ effect on the quality indicators of finished beer. The topic under consideration is relevant since it touches on the beer matrix colloidal stability when compounds with potential destabilizing activity are introduced into it from the outside. Methods: The industrial beer samples’ quality was assessed by industry-accepted methods and using instrumental analysis methods (high-performance liquid chromatography methods—HPLC). The obtained statistical data were processed by the Statistics program (Microsoft Corporation, Redmond, WA, USA, 2006). Results: The study made it possible to make assumptions about the functional dependence of the iso-α-bitter resins and isoxanthohumol content in beer samples. Mathematical analysis indicate interactions between protein molecules and different malted grain and hop compounds are involved in beer structure, in contrast to dry hopped beer, where iso-a-bitter resins, protein, and coloring compounds were significant, with a lower coefficient of determination. The main role of rutin in the descriptor hop bitterness has been established in kettle beer hopping technology, and catechin in dry beer hopping technology, respectively. The important role of soluble nitrogen and β-glucan dextrins in the perception of sensory descriptors of various technologies’ beers, as well as phenolic compounds in relation to the formation of bitterness and astringency of beer of classical technology and cold hopping, has been shown. Conclusions: The obtained mathematical relationships allow predicting the resulting beer quality and also make it possible to create the desired flavor profiles.

## 1. Introduction

The beer’s quality as a colloidal system is determined by the organic compounds that form its structure. The grain (malted and unmalted cultures), the plant raw material (hops and hop products), and the yeast strain ultimately determine the sensory profile of the finished beer [1]. It is confirmed that, on the one hand, the overall complexity of the beer proteome is narrow and is caused by the protein molecules of *Hordeum vulgare* (barley) as a primary source, and yeast, but, on the other hand, there is a large variety of post-translational modifications (PTMs) created as a result of hydrolytic cleavage (proteolysis), glycation, and glycosylation during malting, mashing, and other technological stages of brewing. In particular, it is the protein molecules bound to non-protein components by covalent, ionic, van der Waals interactions that determine the structure and properties of beer [1,2].

Polyphenolic compounds from both grain raw materials and hops are important, in terms of functionality, in relation to beer quality.

Of all the classes of phenolic compounds, grains (rice, barley, wheat, and oats) are rich in phenolic acids, flavonoids, lignans, and tocols [3].

Hops contain phenolic acids, flavan-3-ols, and flavanols, including condensed, such as cereals, and authentic prenylflavanoids (xanthogumol, 6-,8-prenylnaringenin) and stilbenes in their composition [4].

It is reported that the malt (by 70 ÷ 80%) and the hops (by 20 ÷ 30%) contribute to the phenolic profile of the beer [5]. The profile consists of simple phenols; benzoic and cinnamic acid derivatives; coumarins; catechins; di-, tri-, and oligomeric proanthocyanidins; (prenylated) chalcones and flavonoids; as well as alpha- and iso-alpha acids [5]. Their wide range of variations determines the type of intermolecular interactions with protein molecules and the direction of their influence.

The polyphenols of both malt and hop under conditions of kettle hopping were found to optimize the reducing activity and reduce the carbonyl content during the fermentation process; they reduced, in particular, the intensity of the “harsh taste” in fresh beer, and in general, had a positive effect on flavor stability. However, plant polyphenols have been shown to have a negative effect on haze stability [6].

The hop’s phenolic compounds are still under study to this day. It has been shown that hops (*Humulus lupulus* L.) contain several physiologically active polyphenols important both for brewing and for other industries due to their potential, which is determined by genetic factors [7].

Various phenolic compounds, including hops, transferred into the beer affect the taste in general and, in particular, its taste fullness, and cause astringency and colloidal formation: flavonoid oxidation affect astringency, haze, and color, and low-molecular phenols (4-vinylsyringol) can give the beer extraneous aromas during storage (Table 1) [8].

In addition, flavonoids (taxifolin-*o*-glucoside, quercetin-*o*-glucoside, apigenin-6,8-dipentoside, and isofraxidin-*o*-glucoside) and phenolamides were also confirmed in beer [9].

There is a difference between phenolic compounds of malt and hops: despite the small number of phenolic compounds released into wort and beer compared to malt, hop polyphenols are more reactive with respect to protein deposition, which is associated with polarity and the polymerization index (degree of oxidation); in other words, the lower the polymerization index of polyphenols, the more active they are in association with wort proteins [21]. The polarity of phenolic compounds depends directly on the degree of electron cloud coverage of the atoms inside the molecules [22]. Moreover, the phenolic molecule associated with another compound already has a lower polarity than the free form: for example, the polarity coefficient for caffeic acid is (−1.76) and for its methylated compound is (+0.52) [23,24].

Quantitatively and qualitatively, polyphenols influence the composition and equilibrium of the beer matrix, and this influence depends not only on the structure and properties of the compounds themselves but also on external technological factors (temperature, pH, presence of microorganisms, and polar solvents) [25].

There is also an inverse relationship, where the plant matrix compounds affect the configuration of the polyphenols, which affects the sensory perception of beer and the role of polyphenols in this perception. It was found that the perceived bitterness of beer does not stand out in the overall experience of drinking beer, and the drink tastes softer because the bittering compounds (iso-α-resins and phenolic compounds) are associated with glucose residues in the beer matrix [26].

Phenolic compounds of plant objects play an important role. The representatives of phenolic compounds in cereals are phenolic acids, flavonoids, and lignans, both conjugates and aglycones [27]. The distribution of phenolic representatives in the grain structure is determined by their functional significance: phenolic acids (mainly ferulic, *p*-coumaric, and caffeic acids) are in the grain’s cortical layer, and provide antioxidant, antimutagenic effects [28,29]; flavonoids also have functional properties and are pigments of the grain [30]; lignans in connection with other compounds provide mechanical protection of grains from mechanical and other damages [31]. In terms of beer quality, they have an effect when mashing grain products under the condition of the high alkalinity of water. Most of the phenolic compounds in the husks are removed with the grain pellets at the wort filtration stage.

Endosperm polyphenols are more important in terms of their effect on beer quality. The endosperm of cereals contains aglycones form of ferulic, parabiosan, protocatechuic, gallic, and caffeic acids; the other representatives of phenolic compounds are associated with protein, carbohydrate, and other compounds in the whole grain and are released to some extent only during malting [32,33]. The main group of polyphenols in malted grains is the flavone-3-ol group, in particular (+)-catechin, (−)-epicatechin, prodelfinidin B3, and procyanidin B3, as previously reported [34].

In the structure of malt, phenolic compounds often appear as esters, glycosides, and complexes with polysaccharides (arabinoxylans, β-glucans, etc.) [35]. Hop lupulin glands contain a mixture of prenylated, geranylated, oxidized, and/or cyclized chalcones along with bitter acids and volatile oils [36], some of which are in the form of *o*-glycosides that determine their bioavailability [37].

By entering the wort through the process, malt and hop polyphenolic compounds interact with matrix structure-forming compounds (protein, carbohydrate, coloring, and aroma compounds, etc.). Therefore, it is important to consider the other biomolecules of the plant matrix of beer.

Thus, the parameters of the external environment and the hop application moment are important regarding the formation of the beer’s taste profile and its effect on consumer perception. The present study aimed to establish the behavior patterns of hop polyphenolic compounds under dry hopping conditions in combination with other beer matrix compounds.

## 2. Results

### 2.1. The Determination and Mathimatical Analysis of Beer’s Sample Composition

The beer samples studied in this work (Table 5 data) were selected in such a way as to reveal the difference in the influence of various phenolic and other compounds of plant raw materials both in the classical kettle hopping technology and in dry hopping technology on the beer’s flavor descriptors.

Table 2 shows the composition of the tested beer samples.

Based on Table 2 for non-alcoholic beer (1NABK ÷ 3NABK samples), we calculated the multiple regression Equation (1):Y = −158.1691 + 7.6739 · X_1_ − 11.0625 · X_2_ − 24.3165 · X_3_ + 0.2346 · X_4_(1)
where
Y—soluble nitrogen content, mg/L;X_1_—β-glucan content, mg/L;X_2_—iso-α-acids content, mg/L;X_3_—original extract, °P;X_4_—color, °EBC.

The analysis of the multiple regression equation for non-alcoholic beer (1) showed that the most significant relationship exists between the original extract data (the amount of grain raw materials) and the concentration of nitrogenous compounds: with an increase in the amount of raw materials used, the content of soluble nitrogenous compounds are reduced, which can be explained by coagulation processes on the part of the beer matrix and high-molecular protein compounds, unstable due to molecular weight, as well as a decrease in nitrogenous compounds in the process of alcohol removal in non-alcoholic beer technology [38,39,40,41].

However, it is important to note that there is a correlation between changes in nitrogen content and the level of iso-α-bitter hop products: as the level of isohumulone increases, the content of nitrogen compounds decreases, which describes the process of wort clarification during hopping and fermentation [42,43]. Mathematical analysis revealed a strong correlation between the soluble nitrogen and β-glucan (r = 0.941) content; isogumulon content and beer color degrees (r = −0.949), which was confirmed by pair correlation coefficients between the content of nitrogen compounds, β-glucan and isogumulon (r = 0.999); nitrogen content, β-glucan and color degrees (r = −0.992) with the overall system determination coefficient R^2^ = 0.999. In other words, the beer’s plant matrix includes interactions between protein molecules and β-glucan, β-glucan and iso-α-bitter acids, and peptide molecules and reducing compounds, including β-glucans, are involved in Maillard reaction and provide their contribution to beer color.

Based on the data in Table 2 for beer (4ABK ÷ 14ABK samples), we calculated the multiple regression Equation (2):Y = −2628.8934 − 3.0567 · X_1_ − 11.1733 · X_2_ + 324.2603 · X_3_ + 6.3317 · X_4_(2)
where
Y—soluble nitrogen content, mg/L;X_1_—β-glucan content, mg/L;X_2_—iso-α-acids content, mg/L;X_3_—original extract, °P;X_4_—color, °EBC.

The analysis of the multiple regression Equation (2) for alcoholic beer showed that the most significant relationship exists, as well as for non-alcoholic beer, between original extract content (amount of grain raw material) and the concentration of nitrogen compounds; further, we mark the significance of the relationship between changes in nitrogen content and the level of hop products iso-α-bitter resins. The significance of the protein compounds content on the level of β-glucan and the beer’s color dependence has changed compared to Equation (1). In our opinion, in the production of non-alcoholic beer technologies are used to release alcohol that more reduce the concentration of melanoidins, polyphenols (catechins), and caramels compared to the reduction of β-glucan levels [38,39,40,41]. The mathematical analysis also revealed a strong correlation between the soluble nitrogen and β-glucan content (r = 0.940); the content of isohumulon and the beer’s color degree (r = 0.932), which was confirmed by pair correlation coefficients between the nitrogen compounds, β-glucan and isohumulon content (r = 0.918); the content of nitrogen compounds, β-glucan, and color degrees (r = 0.998), with a total system determination coefficient R^2^ = 0.930. However, there is a difference in the regression coefficients in the first and second equations. Thus, in alcoholic beer, the significance between the content of nitrogenous compounds and β-glucan is two times higher compared to non-alcoholic beer, and the relationship between β-glucan content and color’s degree increases by 1.5 times when comparing alcoholic and non-alcoholic beer. The difference in the regression coefficients can be explained by the lack of additional processing of alcoholic beer.

The multiple regression Equation (3), according to Table 2 for samples (15ABD ÷ 20ABD), is as follows:Y = 882.9995 + 3.0568 · X_1_ + 18.0064 · X_2_ − 47.7439 · X_3_ − 46.4638 · X_4_(3)
where
Y—soluble nitrogen content, mg/L;X_1_—β-glucan content, mg/L;X_2_—iso-α-acids content, mg/L;X_3_—original extract, °P;X_4_—color, °EBC.

The analysis of the multiple regression Equation (3) for beer produced by the dry hopping technology showed that the most significant relationship exists, as in the first two cases, between original extract content (amount of grain raw material) and the concentration of nitrogen compounds. The significance of the regression coefficient between the content of nitrogenous compounds and the beer’s color degree has increased compared to the first two dependencies. Next, in terms of significance, we note the correlations between changes in nitrogen content and the level of iso-α-bitter resins of hop products. In our opinion, during the dry hopping beer’s production complex extraction, biotransformation, adsorption of bitter, phenolic, and ester compounds in the presence of yeast take place, which, among other things, affects the compounds that provide color formation of beer (melanoidins, polyphenols, caramels, etc.) [44].

The mathematical analysis also revealed, in the comparison with the first dependencies, the absence of a strong relationship between the soluble nitrogen and β-glucan content (r = 0.351), isohumulon content, and beer’s color degrees (r = 0.0458). This model has weaker correlation between the protein and hop iso-α-acids content (r = 0.578), beer’s color degree and β-glucan content (r = 0.514), and hop iso-α-acids content and beer’s color degree (r = 0.757) compared to alcoholic and non-alcoholic beer. Thus, we mathematically confirmed the influence of the hop compounds extraction degree on the beer’s plant matrix condition, when a relationship between the main biomolecules (proteins, β-glucan, melanoidines ect.) is decreased, with a fall in the total system determination coefficient up to R^2^ = 0.656. The coefficient of determination shows how susceptible the system is to influence from the outside, and in the first two cases, the coefficient of determination was close to 1, while in the case of dry hopping, the system is subject to changing by 44.4%, which confirms the multifactorial transformation of biochemical compounds in the case of adding hop products to beer with microorganism cells.

### 2.2. The Determination and Mathimatical Analysis of the Beer’s Phenolic Compounds Composition

Since the issue of the dry hopping beer’s quality is considered, it is necessary to consider favorable conditions for the extraction of hop polyphenol complex compounds in the presence of polar solvent (ethanol) and yeast cells, taking into account the metabolic transformations of different polyphenols [45,46,47,48], the data are presented in Table 3.

Based on the data in Table 2 and Table 3, the polyphenol groups’ contribution influencing the content of iso-α-acids causing hop bitterness was calculated (Equation (4) for samples 1NABK ÷ 14ABK, Equation (5), for samples 15 ABD ÷ 20ABD):Y = 16.6994 + 3.8339 · X_1_ − 0.4284 · X_2_ − 0.3668 · X_3_ − 1.3108 · X_4_(4)
Y = 20.6848 + 10.4828 · X_1_ − 5.8867 · X_2_ + 0.9791 · X_3_ − 0.2249 · X_4_(5)
where
Y—iso-α-acids content, mg/L;X_1_—isoxanthohumol content, mg/L;X_2_—catechin content, mg/L;X_3_—quercetin content, mg/L;X_4_—rutin content, mg/L.

An analysis of the coefficient’s significance before the variables in the correlation multifunctional relationship (4) showed that isoxanthohumol was associated with isomerized resins in the first place, rutin in the second, and then catechin and quercetin. Paired correlation coefficients showed a moderate relationship between changes in iso-α-acids and isoxanthohumol content, and the partial elasticity coefficient revealed a model pattern in which a 1% change in iso-α-acids would cause a 0.93% change in isoxanthohumol with an overall correlation coefficient R = 0.63 and a coefficient of determination R^2^ = 0.40. The analysis revealed great significance of the private correlation coefficients, that is, a system of three variables—the most significant relationship between the content of isoxanthohumol, catechin, and iso-α-acids (r = 0.727); the content of isoxanthohumol, quercetin, and catechin (r = 0.581); and the content of isoxanthohumol, quercetin, and rutin (r = 0.681). In our opinion, the considered mathematical model describes the complex processes when changing the content of phenolic compounds during fermentation and post-fermentation, because quercetin changes the most [46].

The analysis of Equation (5), i.e., the contribution to hop bitterness of dry hopping beer, is shown as in Equation (4); there is a greater influence of isoxanthohumol content, but further, unlike the previous equation, the contribution during dry hopping is made by catechin content, then quercetin and rutin. Paired correlation coefficients showed a strong relationship between changes in iso-α-acids content and isoxanthohumol (r = 0.86) in contrast to Equation (4), and the partial elasticity coefficient revealed a pattern in the model, in which a 1% change in iso-α-acids content would cause a 1.24% change in isoxanthohumol and a 1.145% change in catechin; the overall correlation coefficient R = 0.91, and the determination coefficient R^2^ = 0.83. The analysis of the partial correlation coefficients showed the same dependencies as in the case of Equation (4): the most significant relationship between the isoxanthohumol, catechin, and iso-α-acids content (r = 0.555); content of isoxanthohumol, quercetin, and catechin (r = 0.490); and the content of isoxanthohumol, quercetin, and rutin (r = 0.661). However, compared to the conditions of the mathematical model (4) in dry hopping beer, 82.55% of the total variability Y is explained by changes in factors Xj, and in kettle hopping beer, this indicator was 40.64%. This is entirely understandable in terms of the smaller impact of technological factors on the hop’s phenolic complex transferred to the beer.

The correlation–regression analysis of the phenolic profile made it possible to describe a model for the influence of hop compounds on beer’s color degree with the kettle hopping samples 1NABK-14ABK (Equation (6)) and the dry hopping samples 15ABD-20ABD (Equation (7)) based on data from Table 2 and Table 3:Y = −2.0123 − 0.3583 · X_1_ + 9.2653 · X_2_ + 1.571 · X_3_ − 0.6998 · X_4_ − 6.5776 · X_5_ + 0.05478 · X_6_ + 0.326 · X_7_(6)
Y = 4.9212 − 0.00531 · X_1_ + 1.4217 · X_2_ + 7.0478 · X_3_ − 1.57 · X_4_ + 0.7082 · X_5_ − 0.03378 · X_6_ − 0.07096 · X_7_(7)
where
Y—color, °EBC;X_1_—total polyphenol content, vg/L;X_2_—isoxanthohumol content, mg/L;X_3_—catechin content, mg/L;X_4_—quercetin content, mg/L;X_5_—rutin content, mg/L;X_6_—soluble nitrogen content, mg/L;X_7_—β-glucan content, mg/L.

The multifactor correlation equation for the beer’s color degree intensity of the studied parameters (6) showed the greatest influence of the isoxanthohumol, rutin, and catechin content on the beer’s color degree intensity with the kettle hopping technology. Paired correlation values showed a strong correlation between beer’s color degree and catechin content (r = 0.756); isoxanthohumol (r = 0.661) and total nitrogen (r = 0.563); isoxanthohumol and catechins (r = 0.723); isoxanthohumol and total nitrogen (r = 0.729); catechins and total nitrogen (r = 0.69); and rutin and β-glucan (r = 0.624). When evaluating the partial correlation coefficients, the analysis revealed a strong relationship between beer’s color degree intensity, isoxanthohumol content and total polyphenols (r = 0.728); color intensity, isoxanthohumol content, and quercetin (r = 0.782); color intensity, isoxanthohumol, and β-glucan content (r = 0.654); color intensity, catechin content, and total polyphenols (r = 0.895); and color intensity, catechin, and quercetin/rutin/nitrogen/β-glucan content (r = 0.777/0.745/0.615/0.754 respectively). It is interesting to note that the content of non-starch polysaccharide is strongly correlated with the amount of isoxanthohumol and catechin (r = 0.785), and this correlation strength is greater than its relationship under the same conditions with quercetin (r = 0.568). The calculation of partial elasticity coefficients showed that the content of total polyphenols of 3.09%, isoxanthohumol of 2.13%, catechin of 2.63%, nitrogen of 2.5%, and β-glucan of 2% would change the beer’s color degree with an increase of 1% with a multiple correlation coefficient R = 0.94 and the coefficient of determination R^2^ = 0.88.

The equation of multiple correlation dependence of beer’s color degree intensity on the studied parameters showed that the most significant parameters are the content of catechins, isoxanthohumol, and quercetin, according to Equation (7). Regarding the pairwise regression coefficients, we can say that the most significant paired correlation had the content of total polyphenols and all indicators (r = 0.511 ÷ 0.895), isoxanthohumol, and catechins (r = 0.713). Partial correlation coefficients revealed links between beer’s color degree intensity, the total content of polyphenols, and the content of soluble nitrogen (r = 0.909); color intensity, the isoxanthohumol and catechin, quercetin, nitrogen content (r = 0.607; r = 0.618; r = 0.796); content of total polyphenols, isoxanthohumol, and all integral parameters of the system (r = (0.796 ÷ 0.946)); the content of the total amount of polyphenols, isoxanthohumol, and the content of protein compounds (r = 0.872); the content of catechins, quercetin, and β-glucan (r = 0.687), the content of isoxanthohumol, quercetin, and nitrogen (r = 0.721) with a total correlation coefficient of R = 1 and a determination coefficient of R^2^ = 1. Elasticity coefficient analysis showed that changing the content of catechin by 5.478%, quercetin by 2.91%, and nitrogen by 2.2% would change the beer’s color degree by 1%.

### 2.3. The Beer’s Samples Polyphenol Complex Effect on Tast Descriptors Intecity Perception

As a result of the descriptor analysis, profilograms of beer samples were obtained during the expert evaluation of beer samples (Figure 1a–c).

Next, the evaluation of the descriptors (Figure 1) was combined with the data in Table 2 and Table 3, and the correlation and regression indices are calculated and presented in Table 4.

The data in Table 4 indicate more complex processes concerning the formation of flavor shades in dry hopping beer’s samples, associated with the interaction of various organic compounds.

## 3. Discussion

Studies have shown that the effect of polyphenolic compounds on beer quality is related to many technological and raw material factors. There was a correlation between the change in nitrogen content, the level of iso-α-bitter hop resins, the content of β-glucan, and beer coloration, and it was 1.5–2 times higher in alcoholic beer compared to non-alcoholic beer.

The formation of the colloidal structure of dry hopping beer’s samples continued more intensively at the fermentation stage in the difference from the kettle hopping beer’s samples, due to the extraction of bitter, phenolic, and essential organic compounds from hop preparations, which is confirmed by other authors [44,49]. Researchers have noted the simultaneous extraction of α-bitter resins and the loss of iso-α-acid during dry-hopping through adsorption on yeast cells and hop preparation particles, the effect of the process temperature, and the pH change of beer in the upward direction [50,51]. The pH shift affects the intensity of beer’s color degree since melanoidins depend on the medium acidity [52], which is confirmed by our research. The binding of protein and bitter acids levels is explained by the direct extraction of hop resins during fermentation and their covalent binding to protein compounds through cysteine sites [53].

It is interesting to note that the form of binding of β-glucan molecules in beer can also occur with protein molecules, as was recently found, through the participation of Ca^2+^ ions, as shown in Figure 2 [54].

With regard to the interaction of glucans and other carbohydrates, there is a mechanism of binding to phenols through hydrogen bonds of hydroxyl groups and carboxyl groups of saccharide dextrins, such as glucans and arabinoxylans, etc., as well as through hydrophobic interactions [55]. This has significance in terms of representing the structure of the colloidal system concerning phenolic compounds, which have sensory, color, and structure-forming contributions in the evaluation of beer quality.

The study revealed a relationship between the iso-α-bitter resins content with isoxanthohumol and rutin to the greatest extent in the kettle hopping beer’s samples, and with isoxanthohumol and catechin in the dry hopping beer’s samples. In our opinion, the localization of the combined presence of bittering resins and phenolic compounds of hops is important [6], as well as the significance of the process of hop addition: hops adding during boiling tends to transform the phenolic complex of both hop and malt compounds more than the addition at the premalted stage; this may explain the importance of rutin in kettle hopping relative to the relationship with bittering resins and catechin in dry hopping conditions [48].

The formation of beer color is known to occur due to the presence of melanoidins, caramels, catechins, and riboflavin [56]. However, a correlation between rutin, catechin, and, among others, isoxanthohumol and beer’s color degree intensity under the kettle hoping technology was found in the study. The fact of preventing the isomerization of xanthohumol and isoxanthohumol in the presence of a prenylflavanoid and color can serve as confirmation of the connection between the isoxanthohumol and the color and the presence of melanoidins, caramels, and reductones of roasted malted and unsalted grain raw materials in the preparation of dark stouts [57,58].

The 2-hydroxyl group in the xanthohumol molecule is considered reactive, which provides its functional value [59].

Under the influence of temperature or oxygen, the 2-OH group is oxidized with the cleavage of the double bond, and xanthohumol is converted into isoxanthohumol, and melanoidins and reductones with their antiradical function prevent the binding of reactive oxygen species with the hydroxyl group of xanthohumol [60].

The catechins level in non-alcoholic beer was at 1.62 ÷ 3.96 mg/L, in alcoholic beer 2.96 ÷ 21.78 mg/L, and in dry hopped beer samples 6.44 ÷ 10.89 mg/L, values consistent with those previously reported [61,62]. We note that the obtained correlations confirm the close relationship between the beer’s color degree index and catechin content, which is associated with the chromophore properties of the molecule [63]. The correlation between catechins and proteins is based on the ability to form protein–phenol complexes as part of the antioxidant action through covalent bonds [64]. The relationship between isoxanthohumol and soluble nitrogen is based on the ability of prenylflavanoid to interact with NADN compounds during the active oxygen-binding reaction during intercellular interaction [61]. The close correlation between isoxanthohumol and catechins is based on the triple bond of isoxanthohumol–protein–catechin running in the colloidal system of beer. The intermolecular interactions of rutin and β-glucan were highlighted by us earlier and confirmed by other authors [55].

The color intensity in dry hopped beer depends on the same parameters as in kettle hopped beer samples, but the most significant were phenolic compounds: catechins, isoxanthohumol, and quercetin according to Equation (7). Since there is an active saturation of fermenting beer with hop compounds (phenolic, essential, and others), the influence of these compounds increases: common polyphenols had the most significant binding force in the evaluation of pair correlation with respect to all indicators.

It is interesting to note the effect of quercetin on the intensity of the beer’s color degree index. Under the presence of a polar extractant (ethyl alcohol digested by microorganisms), quercetin participates in the metabolism of yeast cells and is partially adsorbed on their surface due to mannan–glucan sites [48]. We note that the level of quercetin in the samples of dry hopped beer is slightly higher and is 13.04 ÷ 33.41 mg/L compared to the alcoholic kettle hopped beer—2.46 ÷ 31.02 mg/L (Table 3). Non-alcoholic beer contained the least amount of quercetin—0.75–11.02 mg/L, which indicates the loss of quercetin at the stage of beer release from alcohol.

It is known that quercetin also belongs to the class of antioxidants and prevents oxidation reactions in those biological systems where it is present [62]. Being involved in the processes of oxygen capture in the presence of yeast cells decreases its quantity [48], but this leads to an intensification of microbial metabolism, which leads to a greater accumulation of secondary fermentation products that indirectly affect all organic compounds of beer, including melanoidins, catechins, caramels, and others, leading to color index changes [46,47,65]. On the other hand, nitrogen molecules present in the system bind to molecules of phenolic compounds, which leads to their enlargement and subsidence [26,33,49,51]. These processes are confirmed by the correlation coefficients we have obtained.

Additionally, as a confirmation of sedimentation processes, we can say that the levels of β-glucan content confirm that sedimentation and removal of interacting carbohydrates, polyphenols, and nitrogenous compounds from the colloidal system occurs: the amount of β-glucan dextrins in alcohol beer was in the range 62.0 ÷ 240.5 mg/L, in dry hopped beer 31.0 ÷ 186.2 mg/L, and in waster nitrogen 306.8 ÷ 1185.0 and 560.3 ÷ 1075.0 mg/L, respectively; the level of polyphenols in dry hopped beer was higher—131.2 ÷ 328.0 mg/L, and in kettle hopped beer samples was lower—98.4 ÷ 237.8 mg/L (Table 2). Of course, different raw materials and beer’s origin extract content must be taken into consideration (Table 5).

The evaluation of the colloidal matrix compounds effect on the organoleptic profile of beer samples showed differences (Figure 1, Table 4).

By evaluating the score of the kettle hopped beer descriptors, it was shown that isohumulone, isoxanthohumol, and rutin respond to harmonious bitterness, with isoxanthohumol and rutin correlated with soluble nitrogen content, which equalizes the sensation of bitterness (Table 4). We note that the descriptor describing harmonic bitterness contributes more or less to all phenolic compounds.

The contribution of isomerized forms of α-acids has been widely studied. Thus, iso-α-acids are mainly considered responsible for beer bitterness [66], and isomerization affects the compression of the acyloid ring, which allows the perceived bitterness from isomerized resins [67].

The question of understanding the main compounds responsible for bitterness is still open [67], but the influence of different forms of hop iso-α-acids is still held [68,69]. The phenomenon of cyclization of isomerized acids without oxygen with the participation of protons with the formation of tri- and tetracyclic decomposition products, as well as aldehydes, which contributed to the formation of persistent sharp bitterness [67]. There is an opinion that undesirable tones of bitterness come from the products of autodegradation of isomerized hydroperoxy- and hydroxyl-allo-iso-alpha-acid resins [70]. It has been noted that the same BU units can characterize different levels of sensory perception of bitterness at the raw material level [71], which suggests a complex organization of sensory perception of beer bitterness and confirms our findings.

The harmonic bitterness of dry hopped beer samples (Table 4) differs from the kettle hopped beer one mainly in the phenolic profile (isoxanthohumol, rutin, and quercetin) with the influence of soluble nitrogen and dextrins β-glucan. The importance of organic biomolecules in the perception of the sensory descriptor of harmonic bitterness has not been assessed before, based on the studied literature, but researchers confirm the importance of the grain genetics role in the beer’s taste, which is one of the main sources of protein and carbohydrate compounds [72]. On the other hand, there is evidence that the addition of a pectin solution as a source of carbohydrate smoothed the taste of the non-harmonic profile of the beer [73]. In the perception of harmonic bitterness, therefore, the attributes of the beer’s test fullness (soluble nitrogen and starchy and nonstarchy nature dextrins) have a great influence [74].

In the evaluation of the acute bitterness descriptor, which is usually correlated with the isomerized and biotransformed hop resins complex presence [67,68,69,70], the largest contribution in terms of the correlation of the evaluation with specific indicators were dextrin molecules β-glucan, isoxanthohumol, and rutin (Table 4), which correlates with acute bitterness in dry hopped beer’s samples but with a wider range of phenolic compounds. Carbohydrate molecules serve as buffer systems for the expression of certain descriptors expressing the bitterness of beer in the kettle hopped beer samples. In dry hopping beer, a significant contribution was made to prenylflavanoid and other forms of phenolic compounds, as well as soluble nitrogen (Table 4). This is consistent with the previously obtained data, when it was shown that the conditions of dry hopping contribute to the transfer to the fermented beer maximum amounts of procyanidin B3, catechin, and caffeic acid [75], which may lead to the formation of protein–polyphenolic associates, which it will cause turbidity, as well as isomerization reactions of xanthohumol into isoxanthohumol [14].

In the evaluation of hop bitterness, there is an influence of carbohydrate dextrins in the case of kettle hopped beer’s samples and soluble nitrogen in the case of dry hopping beer, along with the influence of bittering and phenolic compounds, quercetin and rutin, in classical hopping and catechin in cold hopping (Table 4).

Phenolic bitterness is important from the point of view of phenolic profile evaluation, since, as it is known, phenolic compounds are very labile and changes in parameters (O_2_, pH, concentrations of potential associate compounds, temperature, etc.) can cause quantitative and qualitative modifications that affect beer taste profile [12,14,26,75].

For this reason, compounds with antioxidant activity are present in the significant factors of the phenolic bitterness descriptor (Table 4), and only partial correlation factors indicate a broader profile of compounds that form the phenolic oxidation status of beer.

The astringency descriptor is associated in brewers with the content of grain phenols (tannins, catechins) that pass into the wort during mashing [6,12]. The pH of the environment is important for the equilibrium state of these compounds since it is known that the structure at pH closer to alkaline zones leads to the transformation and further degradation of catechins and catechin oligomers, which affects the color, taste, and appearance of beer [21]. It is noted that catechin derivatives were found in beer in the form of (+)catechin and (−)epigallocatechin in low amounts [76,77]. Some data suggest the significance of flavonoid configurations with respect to the sensory properties they present. For example, epicatechin is more bitter and astringent than its chiral isomer catechin [78,79]. The position of the double bond, the identity of the monomeric units, and the introduction of extrinsic radicals equally affect the astringency and bitterness of dimeric or trimeric molecules [80,81], which makes it more understandable that dextrins and nitrogenous molecules participate in the formation of different shades of bitterness and astringency.

On the other hand, mutual suppression of bitterness and sweetness in mixtures has been noted [82,83]. The increase in sweetness or viscosity related to dextrins of carbohydrates of both starchy and nonstarchy nature reduced the intensity of bitterness in vermouth [84], astringency in red wine [85,86,87,88,89].

The astringency descriptor was further divided into acute and residual since the perceptibility of taste shades depends not only on quality but also on human physiology [90].

The sharp astringency was evaluated in terms of the greater contribution of isoxanthohumol, rutin, and catechin and balanced with nitrogenous compounds in the kettle hopped beer, whereas in dry hopped beer, isohumulone and quercetin were added to the same compounds (Table 4), which is justified by the largest amount a variety of phenolic compounds that pass into the beer during fermentation [75].

It is necessary to note that the correlation coefficients indicate the influence on each other with respect to the contribution to the sharp tartness of catechin and soluble nitrogen, as well as isohumulone and catechin, which speaks in favor of the inherent ionic or covalent bond between catechin, soluble nitrogen, and iso-α-acid in the colloidal structure of the classical beer. In dry hopped beer, the greatest influence on the descriptor was exerted by β-glucan with catechin, rutin, quercetin, and soluble nitrogen, and the influence of catechin-isoxanthomol-soluble nitrogen agglomerate was inherent (Table 4).

The residual astringency (Table 4) differed from the sharp one in classic beer by the absence of rutin and catechin, and in dry hopped beer, there were no differences in the connections responsible for this descriptor.

Evaluating the regression coefficients, it can be noted that the combination of bitter resins, prenylflavanoid, and catechin are more responsible for the sharp astringency, and the residual bitterness is caused by catechin, quercetin, and rutin, associated with soluble nitrogen and β-glucan dextrins.

## 4. Materials and Methods

### 4.1. The Research Materials

Samples of filtered pasteurised brewery products were purchased in the retail chain in the amount of five bottles (cans) of each sample and stored in a darkened room with controlled parameters: at a temperature of (15 ± 20) °C and air humidity W ≤ (75 ± 2)% before the study. Beers analysed included 20 samples, whose characteristics are represented in Table 5.

### 4.2. The Research Methods

#### 4.2.1. Chemicals

All reagents and standards were of analytical grade. Quercetin, rutin, isoxantogumol, and phenolic acids standards were from Sigma-Aldrich Chemie GmbH (Taufkirchen, Germany) with a purity of ≥99%. Potassium dihydrogen phosphate (KH_2_PO_4_), acetonitrile, acetic acid, orthophosphoric acid (H_3_PO_4_) were purchased from Galachem (Moscow, Russia).

Sulfuric acid, boric acid, hydrochloric acid (HCl), iron (II) sulfate heptahydrate (FeSO_4_·7H_2_O), methanol, carboxymethylcellulose, ethylenediaminetetraacetic acid (EDTA), ferric ammonium citrate ((NH_4_)_5_[Fe(C_6_H_4_O_7_)_2_]), ammonium hydroxide (NH_4_OH), and isooctane were purchased from the limited liability company “Reatorg” (Moscow, Russia).

Chemicals for determination β-glucan content were purchased in Megazyme Int. (Lesher Place Lansing, MI, USA).

Bidistilled prepared water was used in the determinations.

#### 4.2.2. Determination of Original Extract and Alcohol Content

To determine the original extract and alcohol content, the 2.13.16.1 standard MEBAK^®^ method was used [91].

#### 4.2.3. Determination of Nitrogen Compounds

To determine the common amount of soluble nitrogen, the Kjeldahl method (EBC Method 4.9.3) was used [92].

#### 4.2.4. Determination of the Total Content of Polyphenols

We employed the method for determining the mass concentration of polyphenols (EBC Method 9.9) [61].

#### 4.2.5. Determination of Iso-α-Gumulon Mass Concentration

We employed the method for determining the mass concentration of iso-α-gumulon (EBC Method 9.47) [93].

#### 4.2.6. Determination of Catechin Mass Concentration

The determination of the catechin mass concentration was a high-performance liquid chromatography method with an “Agilent Technologies 1200” LC system (“Agilent Technologys”, Santa Clara, CA, USA) equipped a diode array detector. HPLC equipment was fitted column Supelco C18 150 × 4.6 mm 5 μm (Thermo, Waltham, MA, USA) with wavelength 280 nm. The samples and all standards solutions were injected at a volume of 10 μL in a reversed-phase column at 25 °C. The HPLC mobile phase was prepared as follows. Solution A: 50 mM NH_4_H_2_PO_4_ + 1.0 mL of orthophosphoric acid dissolved in 900 mL of HPLC grade water and the volume was made up to 1000  mL with water, and the solution was filtered through 0.45 μm membrane filter and degassed in a sonicator for 3 min, Solution B: acetonitrile. The mobile phase was run using gradient elution: Solution B: acetonitrile. Mobile phase was run using gradient elution: at the time 1 min 5% B, at the time 10 min 15% B, at the time 10 to 45 min 40% B, at the time 45 to 55 min 98% B, and at the time 55 to 60 min 5% B. The mobile phase flow rate was 1.2 mL/min, and the injection volume was 10 μL [94].

#### 4.2.7. Determination of Quercetin and Rutin Mass Concentration

The determination of the quercetin and rutin mass concentration was a high-performance liquid chromatography method with an “Agilent Technologies 1200” LC system (“Agilent Technologies”, Santa Clara, CA, USA) equipped a diode array detector. HPLC equipment was fitted Luna 5 u C18 (2) 250 × 4.6 mm 5 μm (Phenomenex, Global Headquarters, Madrid Avenue Torrance, CA, USA) column with 290 nm wavelength. The samples and all standards solutions at a volume of 20 μL were injected into a reversed-phase column at 25 °C. The mobile phase was 2% acetic acid solution (A) and acetonitrile solution (B) with the ratio (A:B—70:30). The eluent flow rate was 1.5 mL/min [95].

#### 4.2.8. Determination of Isoxantohumol Mass Concentration

A high-performance liquid chromatography method using “Agilent Technologies 1200” LS system (“Agilent Technologies”, Santa Clara, CA, USA) equipped a diode array detector was applied to determine the isoxantogumol mass concentration. HPLC equipment was fitted Kromasil C18 150 × 4.6 mm 5 μm (Supelco, Santa Clara, CA, USA) column with 290 nm wavelength. The samples and all standards solutions at a volume of 10 μL were injected into a reversed-phase column at 25 °C. The mobile phase was acetonitrile solution (A), water (B) and orthophosphoric acid solution (C) with the ratio (A:B:C—40:60:0.1). The eluent flow rate was 1 mL/min [61].

#### 4.2.9. Determination of the Mass Concentration of β-Glucan

To quantify the mass concentration of β-glucan, the standard fermentation method was used (8.13.1) [96].

#### 4.2.10. Determination of the Beer’s Color

To determine the color of beer, the EBC method (EBC Method 9.6) was used [97].

#### 4.2.11. Organoleptic Evaluation of Beer’s Samples by Descriptors

The organoleptic analysis was performed by a professional group of researchers, consisting of 10 people on a five-point scale according to the characteristic taste descriptors selected. Five points mean a strong descriptor shade, four points a well-developed descriptor shade, two points a slightly visible descriptor shade, and one point a subtle descriptor shade. The results obtained were summarized, and the average score was recorded.

#### 4.2.12. Statistical Analysis

Statistical analysis was performed in five replicates. Descriptive statistics were performed and values are expressed as mean ± standard deviation (SD). In the studies, the Student–Fisher method was used, as a result of which multivariate models of the correlation-regression dependence of the studied parameters were obtained. The reliability limit of the obtained data (*p* ≥ 0.95) was considered to assess various factors affecting the content of polyphenols in all studies; statistical data were processed by the Statistics program (Microsoft Corporation, Redmond, WA, USA, 2006).

## 5. Conclusions

The article examined and evaluated the participation of various organic compounds in the formation of the beer’s quality produced by different hopping technologies. It was shown that the level of the phenolic profile of dry hopped beer is quantitatively higher compared to kettle hopped beer, and its influence on the taste descriptors and beer’s color index of beer’s both types was evaluated. The important role of soluble nitrogen and β-glucan dextrins in the formation of the main beer flavor descriptors as well as the distinctive features of flavor formation was established. For the first time, agglomerates of phenolic and other compounds responsible for shaping the shade of sharp and residiual astringency in the taste of beer are presented. The data obtained will allow a better assessment of the contribution of phenolic compounds and their influence in relation to the creation of desired beer flavor profiles.

## Figures and Tables

**Figure 1 molecules-27-00740-f001:**
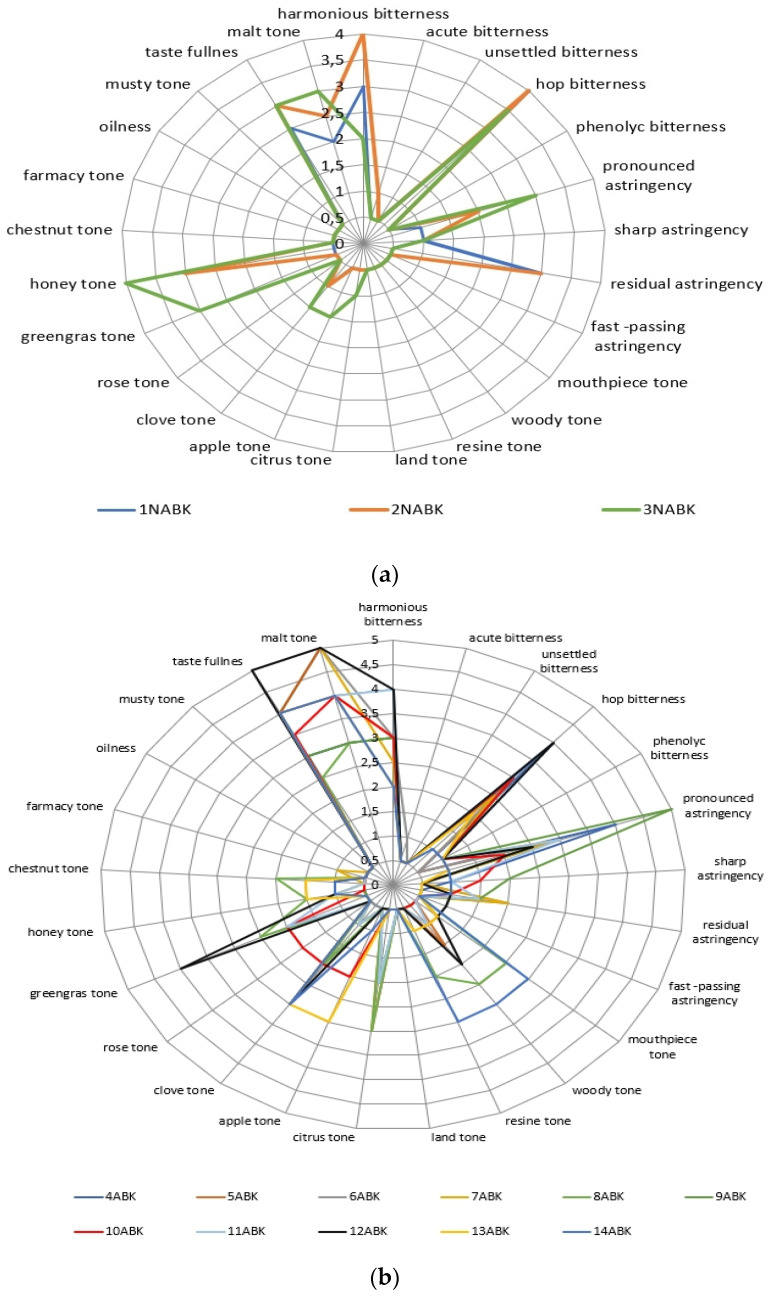
The beer’s sample descriptor analysis data: with kettle hopping non-alcoholic samples (**a**), alcoholic samples (**b**), and dry hopping samples (**c**).

**Figure 2 molecules-27-00740-f002:**
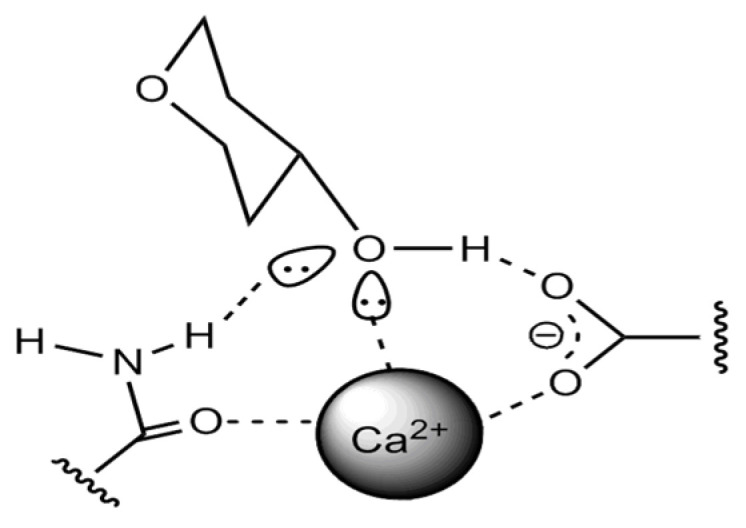
The protein–carbohydrate molecules connection in beer.

**Table 1 molecules-27-00740-t001:** Phenol profile of beer.

Phenol Class/Compound	Associate Compounds	Plant Issue	Teste Contribution	References
Catechins (flavan-3-ols)			bitterness harsh, medicinal, and metallic	[9,10,11,12,13]
(+)-catechin	not associated	cereal/hop		
(−)-epicatechin	not associated	hop		
(+)-catechin	gallic acid	cereal		
(−)-epicatechin	gallic acid, 4′-*O*-Methyl, glucuronic acid	hop		
Proanthocyanidins (condensation products of flavan-3-ol monomers) Procianidins (di-, tri-, and tetra-catechin and epicatechin associated monomers)	gallic acid, 4′-*O*-Methyl, glucuronic acid	hop	bitterness	[9,10,11,12,13]
Prodelfinidins (gallocatec hin, epigallocatechin, and di-, tri-, and tetramers)		cereal		
Flavanones Isoxanthohumol 6- and 8-prenylnaringenin 6-geranylnarin-genin	residual glucose	hop	bitterness	[14]
Flavones apigenin chrysoeriol tricin	residual glucose	cereals	astringency	[15]
Flavonols kaempferol quercetin rutin	residual glucose	cereals hop	bitterness	[16]
Monophenols Gallic acid, protocatechic acid, caffeic acid, vanillic acid, ferulic acid, *p*-coumaric acid, syringic acid, and their aldehydes	bound form as esters, glycosides, and bound complexes	cereal hop	harsh, bitter–sweet, sour, astringent, peppery, medicinal woody, roasted	[17,18,19,20]

**Table 2 molecules-27-00740-t002:** The beer’s samples characteristics.

Sample Number	The Content in Samples, mg/L, Reliability Limit *p* ˂ 0.05					
	Alcohol, vol%	Original Extract, °P	β-Glucan (Gl)	Iso-α-Humulon (IBU) (IH)	Soluble Nitrogen (SN)	Color, °EBC
1NABK	(0.49 ± 0.05) *	7.0 ± 0.70	65.0 ± 4.6	11.8 ± 0.06	300.4 ± 12	6.75 ± 0.20
2NABK	0.48 ± 0.05	7.5 ± 0.70	69.8 ± 4.9	21.6 ± 0.11	439.8 ± 18	5.00 ± 0.15
3NABK	0.48 ± 0.05	7.8 ± 0.80	108.6 ± 7.6	13.5 ± 0.07	630.6 ± 25	7.50 ± 0.22
4ABK	4.6 ± 0.40	10.7 ± 1.00	62.1 ± 4.3	9.7 ± 0.05	459.4 ± 20	5.25 ± 0.16
5ABK	4.5 ± 0.40	10.8 ± 1.00	124.1 ± 8.7	6.3 ± 0.03	445.4 ± 25	5.75 ± 0.17
6ABK	5.1 ± 0.50	11.0 ± 1.00	62.0 ± 4.3	12.9 ± 0.06	984.0 ± 40	7.50 ± 0.22
7ABK	4.7 ± 0.40	11.3 ± 1.00	77.6 ± 5.4	24.4 ± 0.12	1185.0 ± 47	106.3 ± 3.19
8ABK	4.8 ± 0.50	11.6 ± 1.00	62.1 ± 4.3	26.3 ± 0.13	823.6 ± 33	6.75 ± 0.20
9ABK	4.5 ± 0.40	11.8 ± 1.00	75.6 ± 5.3	12.3 ± 0.06	980.0 ± 40	25.0 ± 0.75
10ABK	5.0 ± 0.50	11.9 ± 1.00	128.0 ± 9.0	14.1 ± 0.07	306.8 ± 12	5.25 ± 0.16
11ABK	5.2 ± 0.50	12.0 ± 1.00	120.3 ± 8.4	12.2 ± 0.06	743.0 ± 30	7.25 ± 0.21
12ABK	5.3 ± 0.50	12.8 ± 1.00	240.5 ± 16.8	4.9 ± 0.02	972.1 ± 39	9.50 ± 0.29
13ABK	8.1 ± 0.80	16.5 ± 1.50	93.1 ± 6.5	26.5 ± 0.13	888.0 ± 36	5.25 ± 0.16
14ABK	9.2 ± 0.90	18.6 ± 1.50	96.2 ± 6.7	12.7 ± 0.06	854.4 ± 34	17.5 ± 0.53
15ABD	4.6 ± 0.40	10.0 ± 1.00	31.0 ± 2.2	29.1 ± 0.15	560.3 ± 22	9.50 ± 0.29
16ABD	4.9 ± 0.50	12.0 ± 1.00	93.1 ± 6.5	28.7 ± 0.14	935.7 ± 37	5.25 ± 0.16
17ABD	6.6 ± 0.70	14.5 ± 1.00	186.2 ± 13.0	32.4 ± 0.16	823.6 ± 33	12.5 ± 0.38
18ABD	5.9 ± 0.60	15.0 ± 1.50	155.2 ± 10.9	58.3 ± 0.30	767.6 ± 30	16.5 ± 0.50
19ABD	5.9 ± 0.60	16.0 ± 1.50	74.5 ± 5.2	42.6 ± 0.21	798.4 ± 32	5.25 ± 0.16
20ABD	7.7 ± 0.80	17.5 ± 1.50	108.6 ± 7.6	76.2 ± 0.36	1075.7 ± 43	17.0 ± 0.51

*—each value represents the mean of five independent experiments (±SD).

**Table 3 molecules-27-00740-t003:** The phenol’s profile of beer samples.

Sample Number	The Polyphenol Content in Samples, mg/L, Reliability Limit *p* ˂ 0.05				
	Total	Isoxanthohumol (IXG)	(+)Catechin (Ct)	Quercetin (Qv)	Rutin (Rt)
1NABK	(41.0 ± 3.7) *	1.6 ± 0.02	1.24 ± 0.01	11.02 ± 0.10	3.53 ± 0.03
2NABK	65.6 ± 6.0	1.1 ± 0.01	1.73 ± 0.01	2.78 ± 0.03	5.24 ± 0.05
3NABK	82.0 ± 7.4	2.4 ± 0.02	3.96 ± 0.04	0.75 ± 0.01	6.56 ± 0.07
4ABK	106.6 ± 9.6	2.2 ± 0.02	3.71 ± 0.04	9.90 ± 0.10	3.50 ± 0.04
5ABK	98.4 ± 8.9	3.7 ± 0.04	2.97 ± 0.03	12.84 ± 0.10	7.39 ± 0.07
6ABK	114.8 ± 10.3	4.5 ± 0.04	3.96 ± 0.04	12.09 ± 0.10	7.72 ± 0.08
7ABK	139.4 ± 12.5	7.4 ± 0.07	21.78 ± 0.22	12.58 ± 0.10	6.05 ± 0.06
8ABK	139.4 ± 12.5	3.8 ± 0.04	2.72 ± 0.03	11.94 ± 0.10	8.20 ± 0.08
9ABK	172.2 ± 15.5	5.2 ± 0.05	21.78 ± 0.22	11.98 ± 0.10	7.83 ± 0.08
10ABK	106.6 ± 9.6	3.0 ± 0.03	3.96 ± 0.04	12.83 ± 0.10	6.41 ± 0.06
11ABK	123.0 ± 11.1	4.0 ± 0.04	4.21 ± 0.04	12.71 ± 0.10	8.74 ± 0.09
12ABK	188.6 ± 11.3	2.2 ± 0.02	5.94 ± 0.06	2.46 ± 0.02	12.98 ± 0.13
13ABK	287.0 ± 25.8	6.1 ± 0.06	8.42 ± 0.08	31.02 ± 0.30	1.96 ± 0.02
14ABK	237.8 ± 21.4	3.5 ± 0.04	12.87 ± 0.13	22.10 ± 0.20	1.84 ± 0.02
15ABD	147.6 ± 13.3	4.2 ± 0.04	7.43 ± 0.07	21.55 ± 0.20	2.80 ± 0.03
16ABD	164.0 ± 14.8	3.4 ± 0.03	6.44 ± 0.05	13.04 ± 0.10	13.64 ± 0.14
17ABD	213.2 ± 19.2	4.7 ± 0.05	8.91 ± 0.09	14.57 ± 0.10	4.43 ± 0.04
18ABD	131.2 ± 11.8	4.6 ± 0.05	7.92 ± 0.08	24.20 ± 0.20	2.14 ± 0.02
19ABD	192.7 ± 17.3	5.3 ± 0.05	10.89 ± 0.11	33.41 ± 0.30	2.38 ± 0.02
20ABD	328.0 ± 29.5	9.4 ± 0.10	10.40 ± 0.10	20.20 ± 0.20	4.11 ± 0.04

*—each value represents the mean of five independent experiments (±SD).

**Table 4 molecules-27-00740-t004:** The beer samples correlation–regression indicators.

Indicators	The Beer’s Samples Group	
	Kettle Hoped	Dry Hoped
	**Bitterness** **Descriptor: harmonious bitterness (hb)**	
significant compounds according to the elasticity coefficient (EC)	IH (EC = 0.86); IXG (EC = −0.84); Rt (EC = 0.57)	SN (EC = 1.73); Qv (EC = −1.34); IH (EC = −1.04); Gl (EC = −0.79); IXG (EC = 0.76)
the descriptor influencing factors and their correlation coefficient (Rc)	IXG/SN (Rc = 0.73); IH/Ct (Rc = 0.72); Ct/SN (Rc = 0.69); hb/Rt (Rc = 0.62) IXG/Qv (Rc = 0.58); hb/Qv (Rc = −0.52) hb-Rt/SN (Rc = 0.71); hb-Rt/IXG (Rc = 0.66) hb-Rt/Ct (Rc = 0.64); hb-Rt/Gl (Rc = 0.56) hb-Qv/SN (Rc = 0.50)	IH/IXG (Rc = 0.84); IXG/Ct (Rc = 0.75) IH/SN (Rc = 0.64); IH-IXG/hb (Rc = 0.96) hb-IXG/IH (Rc = 0.89); hb-IH/IXG (Rc = −0.85) IXG-Ct/hb (Rc = 0.77); IH-SN/hb (Rc = 0.71) hb-IXG/Rt (Rc = 0.69); IXG-SN/hb (Rc = 0.67) hb-IXG/Qv (Rc = 0.63); Qv-Gl/hb (Rc = −0.63) Ct-Rt/hb (Rc = −0.63); hb-IXG/Gl (Rc = 0.55)
the general correlation coefficient (GCC)	0.78	1.0
the general determination coefficient (GDC)	0.62	1.0
the unreported compounds contribution, %	38.4	0.0
	**Descriptor: acute bitterness (ab)**	
significant compounds according to the elasticity coefficient (EC)	Gl (EC = −0.57); IXG (EC = −0.54) Rt (EC = 0.52); SN (EC = 0.51)	SN (EC = −4.23); Ct (EC = 2.12) Qv (EC = −1.94); IXG (EC = 1.72) IH (EC = 0.77); Gl (EC = −0.64)
the descriptor influencing factors and their correlation coefficient (Rc)	Ct/SN (Rc = 0.69); IXG/SN (Rc = 0.62) ab/IH (Rc = 0.61); ab-IH/IXG (Rc = 0.54) ab-IH/Qv (Rc = 0.59); ab-IH/Ct (Rc = 0.52) IXG-Ct/Gl (Rc = 0.73); IXG-Ct/IH (Rc = 0.73) IXG-Ct/Rt (Rc = 0.72); IXG-Ct/Qv (Rc = 0.72)	ab/IXG (Rc = 0.72); IH/IXG (Rc = 0.84) IXG/Ct (Rc = 0.74); IXG/SN (Rc = 0.74) IH/SN (Rc = 0.64); IH-IXG/ab (Rc = 0.91) IXG-Ct/ab (Rc = 0.79); IXG-SN/ab (Rc = 0.71) IH-SN/ab (Rc = 0.69)
the general correlation coefficient (GCC)	0.89	1.0
the general determination coefficient (GDC)	0.80	1.0
the unreported compounds contribution, %	19.6	0.0
	**Descriptor: hop bitterness (hb)**	
significant compounds according to the elasticity coefficient (EC)	IXG (EC = 0.59); GL (EC = −0,29) Qv (EC = −0,22); Rt (EC = 0.20) SN (EC = −0.20)	SN (EC = −3.59); IH (EC = −1.23) Gl (EC = −1.13); Ct (EC = −1.02) IXG (EC = −0.92)
the descriptor influencing factors and their correlation coefficient (Rc)	hb/Qv (Rc = 0.74); hb/IXG (Rc = 0.58) IXG/SN (Rc = 0.79); IXG/Ct (Rc = 0.72) IH/Gl (Rc = 0.50); hb-IXG/Rt (Rc = 0.76) hb-Qv/IH (Rc = −0.71); hb-IXG/Gl (Rc = −0.57); hb-IXG/IH (Rc = 0.52)	IH/IXG (Rc = 0.80); IXG/Ct (Rc = 0.77) IXG/SN (Rc = 0.76); IH/SN (Rc = 0.62) hb-IH/IXG (Rc = −0.87); hb-IXG/IH (Rc = 0.81) IXG-Ct/hb (Rc = 0.79); hb-Qv/Gl (Rc = −0.77) hb-IXG/Rt (Rc = 0.61); Qv-Gl/hb (Rc = 0.73) IH-SN/hb (Rc = 0.72); IXG-SN/hb (Rc = 0.71)
the general correlation coefficient (GCC)	0.84	1.0
the general determination coefficient (GDC)	0.70	1.0
the unreported compounds contribution, %	29.9	0.0
	**Descriptor: phenolic bitterness (pb)**	
significant compounds according to the elasticity coefficient (EC)	IXG (EC = −0.55): Qv (EC = 0.52) Rt (EC = 0.55)	SN (EC = 2.52); Qv (EC = −2.31); Gl (EC = −1.24); Rt (EC = −0.80)
the descriptor influencing factors and their correlation coefficient (Rc)	pb/Qv (Rc = 0.43); IXG/SN (Rc = 0.73) IXG/Ct (Rc = 0.72); IXG/Rt (Rc = 0.59) Ct-SN/pb (Rc = 0.64); Rt-Gl/pb (Rc = 0.63) IH-Gl/pb (Rc = −0.59); pb-Qv/Rt (Rc = 0.56); pb-Qv/Gl (Rc = 0.51)	IH/IXG (Rc = 0.78); IH/Ct (Rc = 0.52) IH-IXG/pb (Rc = 0.90); IXG-Ct/pb (Rc = 0.85); pb-IH/IXG (Rc = −0.71); IXG-SN/pb (Rc = 0.70) Qv-Rt/pb (Rc = 0.69); pb-IXG/Ct (Rc = 0.64) IH-SN/pb (Rc = 0.59)
the general correlation coefficient (GCC)	0.81	1.0
the general determination coefficient (GDC)	0.65	1.0
the unreported compounds contribution, %	34.5	0.0
	**Astringency** **Descriptor: pronounced astringency (pa)**	
significant compounds according to the elasticity coefficient (EC)	Gl (EC = −0.58); Rt (EC = 0.55); Ct (EC = 0.50); IXG (EC= −0.46)	SN (EC = −6.93); Ct (EC = 6.00); IH (EC = 3.14) Qv (EC = −2.36); Gl (EC= −1.26); Rt (EC = 1.17)
the descriptor influencing factors and their correlation coefficient (Rc)	pa/Rt (Rc = 0.55); pa-IXG/Rt (Rc = 0.56) pa-Qv/IXG (Rc = −0.64); pa-Rt/Gl (Rc = 0.64); Ct-SN/pa (Rc = 0.67)	pa-IXG/Ct (Rc = 0.89); pa-Qv/IH (Rc = 0.88); pa/Ct (Rc = −0.54); pa/Qv (Rc = −0.74); pa-Ct/SN (Rc = −0.70); IXG-SN/pa (Rc = 0.73) pa-IH/Ct (Rc = 0.69); pa/Rt (Rc = 0.50)
the general correlation coefficient (GCC)	0.85	1.0
the general determination coefficient (GDC)	0.72	1.0
the unreported compounds contribution, %	27.9	0.0
	**Descriptor: sharp astringency (sa)**	
significant compounds according to the elasticity coefficient (EC)	SN (EC = 1.08); IXG (EC = −0.97) Rt (EC = 0.85); Ct (EC = 0.72)	Ct (EC = 7.26); SN (EC = −6.36); IH (EC = 2.34); IXG (EC= −1.61); Qv (EC = −1.78); Rt (EC = 1.30)
the descriptor influencing factors and their correlation coefficient (Rc)	sa-Ct/SN (Rc = 0.68); sa-IXG/Ct (Rc = 0.60) Ct-SN/sa (Rc = 0.83); IH-Ct/sa (Rc = 0.82) Rt-Gl/sa (Rc = 0.62); IH-Gl/sa (Rc = 0.56)	sa/Gl (Rc = 0.77); sa-Gl/IH (Rc = 0.91); sa-Gl/Ct,Qv, Rt,SN (Rc = 0.86); sa-IH/Gl (Rc = −0.80); sa-Gl/IXG (Rc = 0.79); IXG-Ct/sa (Rc = 0.76); IXG-SN/sa (Rc = 0.75)
the general correlation coefficient (GCC)	0.85	1.0
the general determination coefficient (GDC)	0.72	1.0
the unreported compounds contribution, %	28.4	0.0
	**Descriptor: residual astringency (ra)**	
significant compounds according to the elasticity coefficient (EC)	IXG (EC= 0.69); IH (EC = −0.66) Gl (EC = −0.62); SN (EC = 0.53) Qv (EC = −0.52)	SN (EC = 65.56); Ct (EC = −77.84); IH (EC = −33.37); IXG (EC= 23.67); Qv (EC = 26.106); Rt (EC = −11.44)
the descriptor influencing factors and their correlation coefficient (Rc)	IXG-SN/ra (Rc = 0.73); IXG-Ct/ra (Rc = 0.72) IH-IXG/ra (Rc = 0.71); IXG-Qv/ra (Rc = 0.65)	IH-IXG/ra (Rc = 0.83); ra-SN/Gl (Rc = −0.79); Qv-Rt/ra (Rc = −0.77); SN-Gl/ra (Rc = 0.75); ra-IH/Rt (Rc = −0.70); Ct-Qv/ra (Rc = 0.69); IXG-Ct/ra (Rc = 0.68); ra-Rt/Qv (Rc = −0.66); ra-Gl/IH (Rc = 0.65); IH-Rt/ra (Rc = −0.64); Ct-Rt/ra (Rc = −0.64); ra-IH/Ct (Rc = −0.55)
the general correlation coefficient (GCC)	0.61	1.0
the general determination coefficient (GDC)	0.37	1.0
the unreported compounds contribution, %	62.8	0.0

**Table 5 molecules-27-00740-t005:** The beer samples characteristics.

Sample Code	Raw Material List	Yeast Type	Color	Hopping Technology
7 ÷ 8 °P				
1NABK *	light barley malt, hop	lager	light	kettle hopping
2NABK	light barley malt, hop products	lager	light	
3NABK	wheat and barley malts, hop products	ale	light	
11 ÷ 13 °P				
4ABK **	light barley malt, rice, hop, hop products	lager	light	kettle hopping
5ABK	light barley malt, maize, hop, hop products	lager	light	
6ABK	light barley malt, hop	lager	light	
7ABK	light barley and caramel malts, hop and hop products	lager	dark	
8ABK	light barley malt, hop	lager	light	
9ABK	light barley and caramel malts, hop and hop products	lager	dark	
10ABK	light barley malt, hop	lager	light	
11ABK	light barley malt, hop and hop products	lager	light	
12ABK	wheat and barley malts, hop and hop products	ale	light	
14 ÷ 18 °P				
13ABK	dark barley malt, wheat, sugar, hop, hop products	ale	light	kettle hopping
14ABK	dark barley malt, wheat, hop, hop products	ale	dark	
10 ÷ 18 °P (IPA)				
15ABD ***	light barley malt, hop products	lager	light	dry hopping
16ABD	light barley malt, hop products	lager	light	
17ABD	light barley malt, hop products	ale	light	
18ABD	light and caramel, wheat malts, hop products	lager	light	
19ABD	light barley malt, hop	lager	light	
20ABD	light and caramel barley malts, hop products	ale	light	

* NABK—nonalcoholic kettle hopping beer; ** ABK—alcoholic kettle hopping beer; *** ABD—alcoholic dry hopping beer.

## Data Availability

All data are presented only in the materials of the article.

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
