# Peer review of "The Influence of Hop Phenolic Compounds on Dry Hopping Beer Quality"

_molecules, 2022, doi:10.3390/molecules27030740_

Round 1

Reviewer 1 Report

The authors present in their paper the effect of phenolic compounds on beer quality maintaining very accurate measurements and the statistical model provided.

The manuscript is written extremely carefully as well as the quality of the figures is of a very high standard.

The statistical model presented is extremely important for food chemistry. The statistical model presented is extremely important for food chemistry. I believe it is a good start for more extensive research on beer and its constituents, not only phenolic ones.

As a reviewer looking at the complete paper, I find the methodology presented as well as the data shown therein are complete and sufficient to draw conclusions, which the authors present in a good way in the discussion and conclusion sections.

After reading the manuscript, it is very hard to find drawbacks.

Minor revision mode:

However, in my opinion,

A. the literature section should not include tables and graphs. The results shown therein should be properly commented on with literature references.

B. When studying polyphenols, the use of 2% formic acid due to my experience is too much. At this amount, the acid on the column has a destructive effect causing a loss in concentration of the compound. It is standard to use 0.1% formic acid. Have the authors tried using a weaker acid in this case? 

C. What quality of chemical reagents including standards were used for experiments? A reagent paragraph should be provided along with information on the water used and its conductivity.

Author Response

We thank to the reviewer for attention to our publication. Changes made to the article 

Reviewer 2 Report

To authors:

  • The article named “The influence of hop phenolic compounds on dry hopping beer 2 quality” addresses the potential complete determination of beer quality. Overall, all analytical methods are acquired from literature or regulatory agencies. I find a lack of technical development of the entire work.
  • The Abstract is difficult to read. It is suggested to condense sections into a single short paragraph. Also, the abstract needs to be sharp and precise.
  • The introduction provides sufficient background understanding but needs revisions. Please see my comments below.
  • The references are appropriate.
  • Presentation of data/information: Please see my comments below.
  • Figures: Please see my comments below.
  • Overall, this manuscript lacks novelty and the manuscript is not “easy to read”. Currently, this manuscript does not meet the publishing criteria of the Journal.

  1. Line 42: Is it 70-80%? Please check the manuscript for similar errors/typos.
  2. Figures 1 and 2 are not readable. Please use high-resolution images. It is highly suggested to represent this data in a different method and move to the supporting information section. These 2 figures do not add much light to the content in its current form.
  3. What is the objective of the study? What is the novelty of the study? What makes this manuscript unique compared to publications with similar contents?
  4. Table 2 and other results: How do you compare your results with manufacturers or regulators provided numbers?
  5. Figure 3: Please include a high-resolution image. The current version is not readable.
  6. Results section: Highly recommended to rearrange the content into a more readable and interesting format.
  7. Line 552: Did you analyze different lots of beer? My suggestion is analyzing a minimum of 3 lots will reveal variations of the quality and robustness of the analysis.
  8. Line 565 and 566: Please check spellings of alcohol (typo)
  9. Line 580 and 594: particle size must be written as 5 µm

Author Response

The authors thanks the reviewer for his attention to the article 

Reviewer 3 Report

The article deals with the effect of phenolic hop compounds on beer quality indicators. Statistical processing of the collected data makes it possible to predict beer quality and possibly create specific flavour profiles by selecting basic ingredients on the basis of their chemical composition.

The work is well designed and the data presented are sufficient to support the conclusions.
I would like to highlight some minor criticalities:

Abstract:

I don't want to question the results of the statistical analysis but I can't find any reference to the Microsift "Statistics" software. I would insert further references such as the authors, the software version and possibly an indication of where it can be found.

Introduction: 
The paragraphs beginning with line 85 and line 89 contain (in my opinion) unscientific concepts. What is meant by "influence of composition and equilibrium of the beer matrix"? Vague terms such as "structure and properties" or "configuration form of the polyphenols" should be clarified. This is probably jargon for experts in the field which should be explained.

figure 1: 1) enlarge and improve image quality. 2) The caption should be more descriptive (what does 'comulative signal abundance' mean?).
Figure 2: 1) Enlarge and improve image quality. 2) The caption needs to be more descriptive.

Results: 

In the titles of paragraphs 2.1 and 2.2 I would replace the word 'determination' with 'statistical analysis'.

2.1:  1) For the sake of clarity, before presenting the data analysis, I would briefly describe the selected samples and the reason for this selection. 2) I suggest moving the first four paragraphs of the "Results" to the "Introduction" section as the results of the experiment are not discussed.

Materials and Methods
Paragraphs 4.2.5 and 4.5.6 should be revised. e.g.:
- line 578 and 592: The determination of ..... was done with a high-performance liquid chromatography method using an Agilent 1200 series LC system (Agilent Technologies, Santa Clara, CA, USA) equipped with a diode array detector.
- is the Kromasil column Thermo or Supelco?
- "reverse-phase' is not a column but a chromatographic method.

There are also some typos:
line 51: Figure (not Figur)
line 70: Table 1, also replace "Test" with "Taste" in the table heading
line 78: End the sentence after the parenthesis and start again with a new one. 
line 133 Table 2, (not Tabl 2)
line 566: alcohol (not alcogole)

Author Response

All comments of reviewer was accepted, text of manuscript corrected.

The authors thanks to the reviewer for careful attention to the article

Round 2

Reviewer 2 Report

To Authors:

Thank you very much for your careful attention to comments and efforts. After revisions, the content has better readability. The novelty of the work is now elaborated compared to the previous version. Based on my careful evaluation I would like to confirm that the content now meets the publication standards of the journal.